# Physical constraints for respiration in microbial hotspots in soil and their importance for denitrification

Steffen Schlüter[1] , Jan Zawallich[2], Hans-Jörg Vogel[1], Peter Dörsch[3]

[1] Department Soil System Sciences, Helmholtz-Centre for Environmental Research - UFZ, Theodor-Lieser-Str. 4, 06120 Halle, Germany

[2] Institute of Mathematics, TU Clausthal, Erzstr. 1, Clausthal-Zellerfeld, Germany

[3] Faculty of Environmental Sciences and Natural Resource Management, Norwegian University of Life Sciences, NMBU, Aas, Norway

correspondence to: Steffen Schlüter (steffen.schlueter@ufz.de)

**Abstract** Soil denitrification is the most important terrestrial process returning reactive nitrogen to the atmosphere, but remains poorly understood. In upland soils, denitrification occurs in hotspots of enhanced microbial activity, even under well-aerated conditions, and causes harmful emissions of nitric (NO) and nitrous oxide ($N_2O$). Timing and magnitude of such emissions are difficult to predict due to the delicate balance of oxygen ($O_2$) consumption and diffusion in soil. To study how spatial distribution of hotspots affects $O_2$ exchange and denitrification, we embedded microbial hotspots composed of porous glass beads saturated with growing cultures of either *Agrobacterium tumefaciens* (a denitrifier lacking $N_2O$ reductase) or *Paracoccus denitrificans* (a "complete" denitrifier) in different architectures (random vs. layered) in sterile sand that was adjusted to different water saturations (30%, 60%, 90%). Gas kinetics ($O_2$, $CO_2$, NO, $N_2O$ and $N_2$) were measured at high temporal resolution in batch mode. Air connectivity, air distance and air tortuosity were determined by X-ray tomography after the experiment. The hotspot architecture exerted strong control on microbial growth and timing of denitrification at low and intermediate saturations, because the separation distance between the microbial hotspots governed local oxygen supply. Electron flow diverted to denitrification in anoxic hotspot centers

was low (2-7%) but increased markedly (17-27%) at high water saturation. X-ray analysis

revealed that the air phase around most of the hotspots remained connected to the headspace even

at 90% saturation, suggesting that the threshold response of denitrification to soil moisture could

be ascribed to increasing tortuosity of air-filled pores and the distance from the saturated hotspots

to these air-filled pores. Our findings suggest that denitrification and its gaseous product

stoichiometry do not only depend on the amount of microbial hotspots in aerated soil, but also on

their spatial distribution. We demonstrate that combining measurements of microbial activity

with quantitative analysis of diffusion lengths using X-ray tomography provides unprecedented

insights into physical constraints regulating soil microbial respiration in general and

denitrification in particular. This paves the way to using observable soil structural attributes to

predict denitrification and to parameterize models. Further experiments with natural soil

structure, carbon substrates and microbial communities are required to devise and parametrize

denitrification models explicit for microbial hotspots.

## 1. Introduction

Soil carbon and nitrogen turnover is governed by soil heterogeneity at the microscale. Much of

the turnover is concentrated in microsites, providing favorable conditions ($p$O$_2$, temperature, pH)

and substrates (carbon, nutrients) for soil microbial activity. The partitioning of aerobic and

anaerobic respiration in microsites is largely controlled by the water content in the soil matrix

which defines the scale across which O$_2$ diffuses towards microsites of high O$_2$-consuming

activity. Aqueous diffusion lengths range from distances across thin water films in well-aerated

soils, to individual soil aggregates of different radii at field capacity, up to the distance to the soil

surface when the soil is saturated (Smith et al., 2003;Elberling et al., 2011;Ball, 2013;Parkin,

1987). Aerobic respiration is less affected by soil moisture than anaerobic respiration and

typically peaks around water saturations of 20-60% (Schaufler et al., 2010;Ruser et al., 2006;Moyano et al., 2012). Bulk soil respiration starts to decline at higher saturations due to the development of anoxic microsites with lower redox potential, supporting carbon mineralization at typically only a tenth of the rates observed under oxic conditions (Keiluweit et al., 2017). Denitrification, i.e. the dissimilatory respiration of N oxyanions instead of oxygen, is commonly observed at water saturations above 60-70% and peaks beyond 90% (Ruser et al., 2006;Linn and Doran, 1984). The occurrence of anaerobic microsites is governed by the balance between saturation-dependent diffusion and microbial consumption of $O_2$, which in turn depends on the quantity, quality and distribution of soil organic matter in the soil matrix and environmental factors like temperature and pH, which control microbial activity (Tecon and Or, 2017;Nunan, 2017;Smith et al., 2003). In fact, water films around decaying plant material may suffice to induce anaerobic respiration, if microbial respiration exceeds $O_2$ diffusion through that minute barrier (Parkin, 1987;Kravchenko et al., 2017).

The interplay between physical constraints and biological activity in soil controls microbial respiration at microscopic scales and complicates the prediction of denitrification and N-gas fluxes at larger scales. For instance, nitrous oxide ($N_2O$) emissions are notoriously variable in space, which has been attributed to heterogeneous distribution of anoxic microsites in the soil (Mathieu et al., 2006;Röver et al., 1999;Parkin, 1987;Parry et al., 1999). Together with the often observed high temporal variability of microbial respiration and its fluctuations under transient conditions, this has led to the notion of "hotspots" and "hot moments" for microbial activity and emissions (Groffman et al., 2009;Kuzyakov and Blagodatskaya, 2015). "Hotspots" of denitrification have traditionally been linked to diffusion constraints in soil aggregates (Smith, 1980;Arah and Vinten, 1995). Cell numbers and $O_2$ concentration have been shown to decline exponentially towards aggregate centers (Sexstone et al., 1985;Horn et al., 1994;Zausig et al.,

1993;Højberg et al., 1994) and the critical aggregate radius for the development of anoxic centers and is variable but typically >1 mm (Sierra and Renault, 1996;Højberg et al., 1994;Schlüter et al., 2018), However, anoxic microsites have also been reported for smaller aggregates (equivalent diameter of 0.03-0.13 mm) in well-aerated, repacked soils (Keiluweit et al., 2018).

An important, but often neglected aspect of physical diffusion constraints on microbial respiration is the spatial distribution of microbial hotspots within the soil matrix. Incubation experiments were either designed to control the aggregate size in repacked soil (Mangalassery et al., 2013;Miller et al., 2009) or the volume fraction of sieved soil mixed evenly into sterile quartz sand (Keiluweit et al., 2018). Some incubation studies were carried out with undisturbed soil and investigated diffusion constraints within the pore network (Rabot et al., 2015). However, these studies did not address the spatial distribution of hotspots nor the diffusion lengths towards air-filled pores. The vast majority of incubations studies merely reports bulk soil properties like carbon and nitrogen content, bulk density and water saturation. Notable exceptions are Kravchenko et al. (2017) who controlled the position of microbial hotspots by placing decaying plant leaf material into repacked soils with different aggregate sizes and water saturations and Ebrahimi and Or (2018), who placed several layers of remolded aggregates as artificial hotspots into a sand matrix and controlled the volume fraction of anaerobic and aerobic respiration by adjusting the water table in the sand column. Such systematic studies with simplified soil aggregate analogues, yet fully accounting for transport processes from and towards hotspots, including interactions between hotspots, are needed to improve our understanding about how physical constraints on microbial respiration control the anaerobic soil volume and hence denitrification activity.

The objective of the present study was to study the interplay between microbial activity and physical diffusion in controlling aerobic and anaerobic respiration for different spatial distributions of hotspots. We embedded uniform artificial hotspots saturated with oxically growing denitrifier pure cultures (Schlüter et al., 2018) in sterile sand in two different architectures: either densely packed in two layers with minimal distance between hotspots within a layer or distributed randomly with maximum spacing between individual hotspots (~~Figure 1a~~Figure 1b-~~c~~b). We presumed that the competition for oxygen would depend on this separation distance between the hotspots, which in turn would control microbial cell growth and $O_2$ consumption and thus affect the timing and velocity of the aerobe-anaerobe transition in respiration with consequences for the onset of denitrification and its gaseous product stochiometry. Further, by placing hotspots with complete (*P. denitrificans*) and truncated (*A. tumefaciens*) denitrifiers in distinct horizontal layers, we expected to see interactions between hotspots acting as mere $N_2O$ sources and hotspots potentially being $N_2O$ sinks (by reducing $N_2O$ to $N_2$). To create contrasting velocities of oxygen transfer between headspace and hotspots, we incubated the different hotspot architectures at three water saturations thereby addressing the question how overall oxygen supply (regulated by saturation) and local oxygen availability (regulated by hotspot distribution) interact in regulating denitrification activity. The overall goal of our reductionist approach with artificial hotspots, pure bacterial strains and a closed incubation system was to reduce the inherent complexity of soil in order to (i) identify processes that governed denitrification at the pore scale and (ii) create an experimental dataset for validation of pore-scale denitrification models.

For this, we monitored $O_2$, $CO_2$, NO, $N_2O$ and $N_2$ at high temporal resolution and determined the morphology of the air-filled pore space in terms of air connectivity, air tortuosity and air distance by X-ray computed tomography after the experiment.

## 2. Material and methods

### 2.1. Microbial hotspots

Two denitrifier strains were used in this study: *Paracoccus denitrificans* expresses all denitrification enzymes necessary to reduce $NO_3^-$ to $N_2$, whereas *Agrobacterium tumefaciens* lacks the gene *nos*Z encoding nitrous oxide reductase ($N_2OR$), which makes $N_2O$ the final denitrification product. Moreover, the two strains differ in their regulatory phenotypes with respect to inducing denitrification in response to oxygen depletion, which leads to characteristic

patterns of product accumulation (Bergaust et al., 2011). *P. denitrificans* induces NO and $N_2O$ reductase early during $O_2$ depletion (Bergaust et al., 2010), thus releasing little $N_2O$. By contrast, *A. tumefaciens* is known to be less stringent in controlling intermediates, which may result in the release of large amounts of NO, up to cell-toxic, milli-molar concentrations (Bergaust et al., 2008). Both strains were grown oxically in Sistrom's medium (Sistrom, 1960) as described in a

previous study (Schlüter et al., 2018), but at double strength to provide enough substrate for depleting $O_2$ during aerobic growth once transferred to the porous glass beads. The medium was amended with 10 mM $NH_4NO_3$ and 5 mM $KNO_3$ for anaerobic growth. To produce microbial hotspots, porous borosilicate glass beads (VitraPOR P100, ROBU Glasfilter Geräte GmbH) with a diameter of 7 mm, a porosity of 32% and a medium pore diameter of 60 µm were saturated with

freshly inoculated growth medium ($\approx 10^8$ cells ml$^{-1}$) by submersion into one of the two cultures. In the following, the inoculated porous glass beads are referred to as *At-* (*A. tumefaciens*) and *Pd-* (*P. denitrificans*) hotspots. Detailed information about the culture conditions and the inoculation procedure can be found in Schlüter et al. (2018).

## 2.2.    Repacked sand

Fifty *At* and *Pd* hotspots each were placed into 120 ml of washed, sterile quartz sand (0.2-0.5 mm grain size) yielding a volume fraction of 14% (20 ml; Fig. S1a). The sand was packed into 240 ml glass jars (Ball Corporation, Bloomfield, CA) in portions of 10 ml layers and adjusted to target saturation by adding sterile water with a spray can. The packing procedure resulted in some minor changes in porosity between layers and some larger gaps around the hotspots (Fig. S3)

which affected air distribution in the sand (Fig. S2a). Three saturations were used, corresponding to water-filled pore spaces (WFPS) of 30, 60 and 90%. The fully saturated hotspots were placed into the sand at three different architectures (**Fig. 1**). For the "random" distribution, the hotspots were placed in five equidistant (~9.8 mm, center to center) horizontal layers with a random distribution of ten *At* and ten *Pd* hotspots per layer. The average spacing between neighboring

hotspots (surface to surface) was approx. 6mm in this architecture, which was the maximum possible spacing for this amount of hotspot volume (Fig. 1b). For the "layered *At/Pd*" and "layered *Pd/At*" distributions, all fifty hotspots of each strain were placed into one of two horizontal layers spaced 21 mm from each other (center to center) at an average headspace distance of 18.2 and 39.2 mm, respectively, where the order represents *top/bottom*. The average

spacing between neighboring hotspots (surface to surface) was approx. 1 mm in these densely packed layers (Fig. 1c). Care was taken to keep the hotspots cool (on crushed ice) during the packing procedure. The pore size distribution of the porous hotspots and the sand in the bulk soil and in hotspot vicinity are reported in Figure S3.

## 2.3.    Incubation

To establish aerobic and anaerobic growth patterns and denitrification kinetics for both bacterial strains when growing inside the porous glass beads, a pre-experiment was conducted without

sand. Fifty *Pd* or *At* hotspots were placed in septum-sealed 120 ml serum bottles (Fig. S1b) and

incubated at 15°C under either oxic (He/$O_2$ 80/20% ) or anoxic (He 100%) conditions in two

replicates per treatment. Headspace concentrations of $O_2$, $CO_2$, NO, $N_2O$ and $N_2$ were measured

every 4 h by piercing the septum with a hypodermic needle mounted to the robotic arm of an

autosampler (GC-PAL, CTC Analytics, Switzerland). The autosampler was connected to a gas

chromatograph (Agilent Model 7890A, Santa Clara, CA, USA) and a NO analyzer (Teledyne

200. San Diego, CA, USA) via a peristaltic pump. Detailed information about the robotized

incubation system and the experimental setup can be retrieved elsewhere (Molstad et al.,

2007;Schlüter et al., 2018).

In the main experiment, freshly inoculated glass beads were packed into incubation vessels as

described above, three replicates for each of the nine combinations of saturation and hotspot

distribution. Jars with 30% and 60% WFPS were flushed with He/$O_2$ for 40 min, using ten cycles

of vacuum (3 min) and purging (1 min). Jars with 90% WFPS were flushed using 180 cycles of

mild vacuum (~ 600 mbar) and $O_2$/He purging to avoid structural changes of the packed columns

due to bubbling of trapped gas. The jars were then placed into a water bath kept at 15°C and after

temperature equilibration $O_2$/He overpressure was released. Gas concentrations in the headspace

were analyzed as described above. Gas production and consumption kinetics were used to

calculate the fraction of electrons diverted to $O_2$ or N oxyanions and thus to estimate the

contribution of denitrification to total respiration ($e^-_{denit}/e^-_{total}$) (Schlüter et al., 2018;Bergaust et

al., 2011). The NO/(NO+ $N_2O$ + $N_2$) and $N_2O$/(NO+ $N_2O$ + $N_2$) product ratios were estimated

from the cumulative release of gaseous denitrification products (NO, $N_2O$, $N_2$), after subtracting

precursors from products (NO from $N_2O$ + $N_2$ and NO + $N_2O$ from $N_2$). The rationale behind the

latter was to mimic an open system, in which N-gases released to the atmosphere are not

available any longer as electron acceptors for denitrification. This approach cannot make up for

the general shortcoming of closed systems that accumulate gaseous and dissolved intermediates to high concentrations, but $N_2O$ concentrations in soil air reaching >100 ppm are reported occasionally (Risk et al., 2014;Russenes et al., 2019). Details about the calculation of denitrification product ratios can be found in the Supporting Information (SI 1.2).

## 2.4.     X-ray tomography and image analysis

After the incubation experiment, the glass jars were scanned with X-ray micro-tomography (X-tek XCT 225, Nikon Metrology) with a beam energy of 145 kV, a beam current of 280 µA, an exposure time of 708 ms per frame, a 0.5 mm copper filter for reducing beam hardening artefacts and a total of 3000 projection for a full scan. Individual hotspots were also scanned (100 kV,
90µA, 1000ms per frame, no filter) to analyze the internal pore morphology. The 2D projections were reconstructed into a 3D image with a resolution of 35 µm using a filtered-back projection algorithm in the X-tek CT Pro 3D software. Image processing from raw gray-scale data (**Fig. 1a**) to segmented data including sand grains, air and water (**Fig. 1b-c**) was carried out according to well-established protocols for multi-phase segmentation (Schlüter et al., 2014). The porous glass
beads were assigned to *At* or *Pd* hotspots according to the orientation of the flat end in the random architecture or by the vertical position in the layered architecture (**Fig. 1b-c**). The segmented images were analyzed with respect to three different spatial attributes of the air-filled pore spaces deemed important for oxygen supply. 1. Air connectivity by distinguishing isolated air-filled pores and air-filled pores with a continuous path to the headspace (yellow and red in
**Fig. 1d**). Air connectivity is then defined as the ratio of connected air-filled pore space and total air-filled pore space 2. Air tortuosity as derived from the geodesic length of connected air-filled pores. The geodesic length is the distance of any connected air voxel to the headspace along curved paths around obstacles like solid particles and water-blocked pores (**Fig. 1e**). Air

tortuosity is the ratio between geodesic and vertical Euclidean distance to the headspace averaged over all connected, air-filled voxels. It is a proxy for the diffusive transport of gaseous oxygen in air-filled pores 3. Air distances of water-filled pores as defined by the average geodesic distance from any water voxel to the closest air-filled pore with headspace connection (white in **Fig. 1f**). Air distance is a proxy for the slow diffusive transport of dissolved oxygen. All image processing steps were carried out with Fiji/ImageJ (Schindelin et al., 2012) and associated plugins (Legland et al., 2016;Doube et al., 2010) or with VG Studio Max 2.1 (Volume Graphics). Each image processing and analysis step is explained in detail in the supporting information (SI 1.3).

[Figure 1]

## 3. Results

### 3.1. Aerobic respiration and denitrification in unconstrained hotspots without sand

*At* grew faster than *Pd* at 15°C in the experiment with loosely placed porous glass beads as indicated by faster $O_2$ consumption and $CO_2$ accumulation in the oxic treatment (**Figure 2a,b**). Also under fully anoxic conditions, *At* accumulated $CO_2$ faster than *Pd* (Figure 2b). N-gas kinetics clearly reflected the disparate regulatory phenotypes of the two bacterial denitrifiers. Anoxic *At* instantly accumulated large amounts of NO (**Figure 2c**) which persisted until all $NO_3^-$ was reduced to $N_2O$ (as judged from the stable plateau in $N_2O$, **Figure 2d**). Due to slower growth and $O_2$ consumption, *Pd* induced denitrification much later than *At*, but accumulated less intermediates (NO, $N_2O$) than *At*. Oxically incubated *Pd* accumulated no detectable NO, indicating efficient regulation of denitrification when switched slowly to anaerobic conditions in hotspots. Also, NO may have been reduced to $N_2O$ when diffusing from the anoxic center to the

boundary of the hotspots. In the initially oxic treatments, denitrification contributed 7% to the total electron flow in *At* hotspots and 13% in *Pd* hotspots measured over the entire period, reflecting the fact that (i) *Pd* has one more reduction step in the denitrification sequence and that (ii) *At* used less nitrate for anaerobic respiration in anoxic hotspots centers and more oxygen for
aerobic respiration in oxic hotspots margins than *Pd*.

[Figure 2]

## 3.2.       Effects of hotspot distribution in sand

The distribution of microbial hotspots within the sand strongly impacted bulk respiration. This is evident for treatments with medium saturation (60% WFPS) for the first 210 h of incubation
(**Figure 3)** and with other saturations for the entire incubation period (300 h; Figures S4-6). The random distribution of hotspots allowed for much faster aerobic growth than the layered architectures, leading to complete consumption of $O_2$ from the jars within 70 h (**Figure 3a**). Given the slow growth of *Pd* (**Figure 2a**), initial $O_2$ consumption was dominated by the activity of *At* hotspots turning them partly anoxic. Hence, the pronounced NO peak in the random
treatment, coinciding with complete $O_2$ exhaustion from the headspace (**Figure 3c**), was due to *At* denitrification, similar to what was seen in the unconstrained *At* hotspots under anoxic conditions (**Figure 2c**). $N_2O$ production was observed long before $O_2$ was depleted from the headspace (**Figure 3d**) and is attributed entirely to *At* denitrification. *Pd* denitrification did not start before all $O_2$ was depleted and manifested itself in a transient increase in $N_2O$ production at ~70 h
together with an exponential increase in $N_2$ production (**Figure 3e**) which was also observed with unconstrained *Pd* hotspots (**Figure 2e**). Note that the apparent net consumption of $CO_2$ (**Figure 3b**) upon $O_2$ depletion was due to internal alkalization driven by accelerating denitrification, once all hotspots turned anoxic.

[Figure 3]

In the layered architectures, $O_2$ consumption was slower and complete anoxia was not reached before 120 h into the incubation. In contrast to the random architecture, less $O_2$ was available for each individual hotspot in the densely packed hotspot layers, allowing for less aerobic growth per unit time. As a consequence, there was more time for fully denitrifying *At* hotspots to interact with *Pd* hotspots which induced denitrification gradually between 80 and 120 h. Indeed, less $N_2O$

accumulated in the headspace than in the random treatment (**Figure 3d**, S6d) and the onset of $N_2$ accumulation appeared long before complete $O_2$ depletion in the headspace (**Figure 3a,e**). In other words, *Pd* hotspots consumed $N_2O$ produced in *At* hotspots. Upon $O_2$ depletion in the headspace, a burst of NO production occurred (**Figure 3c**) as seen previously with *At* hotspots (**Figure 2c**). However, since *Pd* denitrification was now fully developed, the NO peak was much

more short-lived than with the random distribution, because *Pd* hotspots reduced NO produced by *At* hotspots all the way to $N_2$.

The effect of vertical order in the layered hotspot architecture was small, but consistent among all denitrification products. The distribution with *Pd* hotspots on top (layered *Pd/At*) consumed the NO and $N_2O$ produced in *At* hotspots much quicker than the *At/Pd* architecture (**Figure 3c-d**) and

270 accumulated $N_2$ faster after complete $O_2$ depletion (**Figure 3e**). Both observations highlight the effect of shorter diffusion distances between the headspace and the *Pd* hotspot layer in the layered *Pd/At* architecture.

## 3.3.    Effects of matrix saturation

Differences in water saturation resulted in different absolute amounts of oxygen initially present

in the jars (**Figure 4a**) but did not affect the $O_2$ concentration in the sand matrix. Oxygen was

depleted slightly faster at 60% than at 90% saturation even though there was absolutely more $O_2$ initially present at 60% WFPS. This illustrates the paramount role of oxic growth for the oxic-anoxic transition in the hotspots: the more $O_2$ available initially, the stronger the aerobic growth and the faster the oxic-anoxic transition.

Increasing saturation from 60 to 90% in the randomly distributed hotspots had a strong effect on the timing and accumulation of denitrification products. The expected NO burst upon $O_2$ depletion was damped by two orders of magnitude (**Figure 4c**), because the oxic-anoxic transition proceeded more smoothly in the 90% treatment and NO was reduced further to $N_2O$ before it could escape to the headspace. On the other hand, $N_2O$ and $N_2$ production commenced

earlier in the 90% than in the 60% treatment (**Figure 4d-e**), indicating that $O_2$ availability was *a priori* smaller irrespective of metabolic activity (which was larger in the 60% treatment). The switch from net $N_2O$ production to net $N_2O$ consumption indicates the moment when microbial activity in *Pd* hotspots caught up with *At* hotspots.

     [Figure 4]

Surprisingly, $O_2$ consumption in the 30% treatments was slow despite having the largest amount of $O_2$ in the jar. This was caused by unintended substrate limitation. Due to overlapping pore size distribution between porous hotspots and sand (Fig. S3c), medium was sucked by capillary force from the hotspot into the surrounding sand, as could be seen in a parallel experiment with brilliant blue dye (Fig. S7). This separated bacterial cells, which were likely immobilized in the pore

space of the hotspots, temporarily from a considerable fraction of the carbon and $NO_3^-$ supplied with the medium, before the dissolved substrate would diffuse back into the hotspots due to the evolving gradient induced by consumption in the hotspots. Decreasing the saturation from 60% to 30% also resulted in different timing and accumulation of denitrification products. The slow oxic

growth of both *At* and *Pd* hotspots due to the substrate diffusion limitation at 30% WFPS

provided more time for *Pd* hotspots to interact with *At* hotspots than in the 60% WFPS treatment. Indeed, the NO burst from *At* hotspots after complete $O_2$ exhaustion in the random architecture was 50% higher at 30% WFPS indicating higher *At* cell numbers due to prolonged oxic growth (**Figure 4c,**), whereas the $N_2O$ peak was 50% lower, due to concomitant $N_2O$ reduction in *Pt* hotspots (**Figure 4d**).

### 3.4.   Mass balances

By the end of the incubation, oxygen was exhausted in all treatments. Likewise, $NO_3^-$ was consumed by all treatments, except for the layered hotspots at 30% and 60% WFPS. This means that respiration was electron acceptor limited and that the cumulated recovery of denitrification products can be compared with the amount of $NO_3^-$ initially present (Figure S8). The balance

between aerobic and anaerobic respiration, $e_{denit}^-/e_{total}^-$ (Bergaust et al., 2011), is given by the electron flow to nitrogenous electron acceptors relative to the total electron flow, including $O_2$ respiration (**Figure 5**). When seen over all three water saturations, early stage denitrification under oxic headspace conditions (**Figure 5a**) showed a threshold response to increasing moisture with disproportionally higher $e_{denit}^-/e_{total}^-$ ratios at 90% WFPS (17-27%) than at 60% or 30%.

The proportions of electrons diverted to denitrification at low and medium saturations were small (2-7%) and even smaller than those observed in unconstrained hotspots (7-13%). Differences between saturations were less pronounced when the entire incubation period is considered (**Figure 5b**), since fully anoxic conditions during late stage incubation overrode saturation effects. Overall, the effect of hotspot architecture on $e_{denit}^-/e_{total}^-$ ratios was smaller than the

effect of saturation.

This stands in stark contrast to the pronounced effect of hotspot architecture on denitrification product ratios (**Figure 5c, d**). Hotspot architecture governed growth rates through local competition for $O_2$ and therewith the number of active cells involved in net production sites (*At* hotspots) and net consumption sites (*Pd* hotspots) of NO once $O_2$ was exhausted. In layered

hotspot architectures there was hardly any net-release of NO to the headspace irrespective of saturation (**Figure 5c**). With random hotspot architecture, there was substantial NO release, the magnitude of which, however, decreased linearly with saturation. This pattern in NO stoichiometry clearly reflects the number of *At* cells at the moment of complete $O_2$ depletion, as affected by oxic growth which lasted longer with lower saturation. The $N_2O$ product ratio (**Figure**

**5d**) was influenced by both saturation and hotspot architecture. In layered architectures, the $N_2O$ ratio increased exponentially with increasing saturation similar to what was observed for relative electron flow to denitrification (**Figure 5a**). In random architectures, the $N_2O$ product ratio was consistently higher than in layered architectures irrespective of saturation, yet the highest ratio was reached at 60% WFPS, due to the most vigorous growth, and hence fastest oxic-anoxic

transition at intermediate saturation.

[Figure 5]

## 3.5.    Pore space properties

At the lowest saturation (30% WFPS), the entire air-filled pore space was connected to the headspace (**Figure 6a**) and tortuosity was close to unity, i.e. the diffusion lengths in air only

depended on the vertical distance to the headspace (**Figure 6b**). The diffusion distances in water-filled pores (**Figure 6c**) corresponded to the size of small, evenly distributed water clusters. At medium saturation (60% WFPS), the amount of disconnected air was still negligible and tortuosity only slightly increased. The increase in air distance was due to a few large water

pockets, which were caused by the step-wise addition of water to the repacked sand. Only at 90% saturation a considerable air volume of 5-20% became disconnected from the headspace. The path along which the remaining air was connected to the headspace became more tortuous with increasing saturation and average diffusion distances in water to the connected air cluster increased to 1 mm. This is still surprisingly short as compared to the size of the hotspots (7 mm). Independent tests showed that the high air connectivity at this low air content was facilitated by vacuum application during $He/O_2$-purging prior to the incubation. Directly after packing, the continuous air cluster only reached 10-15 mm into the sand (data not shown), whereas bubbling due to vacuum application formed continuous air channels that reached deep into the sand matrix connecting even the deepest hotspots with the headspace. Moreover, some larger gaps remained around hotspots during packing which tended to be air-filled after wetting. This is reflected in the consistently higher air-connectivity, lower air tortuosity and lower air distance, when only pores in the direct vicinity of hotspots are analyzed (**Figure 6a-c**). More than 90% of hotspot surfaces still had a direct air-filled connection with the headspace at 90% WFPS (**Figure 6a**). Depth profiles of these pore space attributes are reported in Fig. S2.

[Figure 6]

# 4. Discussion

## 4.1. Physical constraints on denitrification kinetics

The experimental setup in this incubation study was designed to investigate physical constraints on microbial respiration in hotspots as affected by the interplay between gaseous diffusion through a sterile matrix and local competition for oxygen. For this, we compared different combinations of water saturation in the matrix and spatial distributions of hotspots, which led to

different physical constraints for the supply of hotspots with oxygen. As a consequence, oxic growth rates differed among treatments which had various implications on denitrification as summarized in a conceptual scheme (Figure 7).

[Figure 7]

Our setup is a coarse simplification of soil in which metabolic activity in hotspots not only depends on oxygen supply, but also on diffusion of substrates from the matrix to the hotspots. As such, our experiment does not allow to draw direct conclusions about the functioning of hotspots in real soils. However, by placing denitrifiers and their substrates into hotspots, we considerably reduced the level of complexity and created a system that is amenable to studying the dynamic

interrelations between denitrifier growth, oxygen consumption and induction of denitrification by gas kinetics.

Soil $N_2O$ emissions are known to be highly variable in time and a unifying concept incorporating dynamic changes in denitrification activity and product stoichiometry in response to changing environmental conditions is still missing. Our model system provides a first data set for

validating mathematical process models that are explicit for structural distribution of hotspots and dynamic changes in boundary conditions (here mimicked by different hotspot architectures and declining oxygen concentrations in the headspace during batch incubation, respectively). The development of such models is a core activity of the DASIM project (http://www.dasim.net/). By combining metabolic measurements with advanced structural imaging and computation, we also

provide a link to parameterizing such models with real soil data in future research.

Inoculating growing denitrifiers into porous glass beads and embedding them in sterile sand resulted in a highly dynamic system with respect to oxygen consumption and induction of denitrification. This was intended for the sake of experimental depth, but it must be noted that

oxic-anoxic transitions are likely slower, i.e. less dynamic in real-soil hotspots. In real soils, even

highly organic hotspots contain a fair amount of recalcitrant organic C that limits microbial

growth and oxygen consumption. Also with respect to denitrification stoichiometry, real soils

may be expected to be less dynamic as multiple denitrifying phenotypes contained in the natural

soil microbiome (Roco et al., 2017) utilize denitrification intermediates mutually.

Notwithstanding, soil NO and $N_2O$ emissions are known to be episodic in nature. Large,

denitrification driven emission pulses occur upon abrupt changes in $O_2$ availability, caused by

external factors like heavy rainfalls or soil freezing (Flessa et al., 1995), $O_2$ consumption by

nitrification after ammoniacal fertilization (Huang et al., 2014) or incorporation of easily

degradable organic matter (Flessa et al., 1995) which cannot be captured satisfactorily by

common steady-state models for soil respiration and $N_2O$ emission (Parton et al., 2001;Li et al.,

1992). Even though the concept of hotspots is central in the understanding of denitrification

dynamics in upland soils, common soil denitrification models do not account for the dynamics of

spatially explicit hotspots in the soil matrix but rather scale bulk denitrification with a generic

anoxic volume fraction (Li et al., 2000;Blagodatsky et al., 2011). To advance soil denitrification

models, it is obvious that microbial respiration dynamics in hotspots have to be targeted, both

conceptually (Wang et al., 2019) and experimentally (Kravchenko et al., 2017;Ebrahimi and Or,

2018). Our study is a first step in this direction.

One of the main findings of this study is that soil microbial respiration and the propensity to

develop denitrifying anoxic hotspots does depend on their distribution in space. The onset of

denitrification and its kinetics were linked to the spatial and temporal extent of anoxia developing

in hotspot centers, which was governed by the interplay between denitrifier growth, diffusional

constraints and hotspot architecture. When distributed randomly, microbial activity was most

disperse relative to available oxygen, resulting in more growth, faster $O_2$ draw down and earlier

anoxia than when packed densely in layers (**Figure 3**). Rapid oxic-anoxic transition led to higher release rates of denitrification intermediates and increased NO and $N_2O$ product ratios (**Figure 5c-d**). This effect was most pronounced at low and intermediate saturations but was dampened at 90%WPFS because oxygen supply was impeded by bulk diffusion irrespective of hotspot placement. Thus, our results highlight the significance of hotspot distribution at low soil moistures and exemplifies why $N_2O$ emissions are notoriously difficult to predict under these conditions.

Even though we failed to fully synchronize *At* and *Pd* growth in time, our experiment demonstrates that contrasting denitrification phenotypes may interact in modulating $N_2O$ flux to the atmosphere. *Pd* hotspots reduced $N_2O$ released from *At* hotspots irrespective of the layers' orientation (**Figure 3d**), which can be attributed to the high degree of air connectivity in the sand column (**Figure 1d**). We had expected more $N_2O$ reduction with *Pd* on top (layered *Pd/At*), but since *At* grew faster than *Pd*, partial anoxia and NO and $N_2O$ formation was induced in *At*, long before $N_2O$ consuming activity was induced in *Pd* hotspots. Future experiments with artificial hotspots should therefore carefully consider potential growth rates and air connectivity in packed soil.

## 4.2.     **Physical constraints on cumulative denitrification**

The cumulative release of gaseous denitrification products, as described by electron flow ratios, depended less on hotspot architecture than on soil moisture. Electron flows to denitrification ranged from <5% of total respiratory flow at low to medium saturations (30, 60% WFPS) to almost 23% at 90% WFPS (**Figure 5a**). We attribute the generally low denitrification electron flow in our experiments to the small active volume relative to the sterile sand matrix (the total volume fraction of hotspots was 14%, less of which was actually anoxic) and the large amount of

oxygen initially present in the incubation jars. Yet, we found a typical, non-linear denitrification response to soil moisture (**Figure 5a**). This threshold behavior is well known (Weier et al., 1993) and has been attributed to a disproportional contribution of small pores to the anoxic volume at higher saturation (Schurgers et al., 2006). In our system, consisting of coarse sand with a

440 relatively homogenous pore size distribution, we attribute the non-linear response to an increase in tortuosity of air-filled pores and an increase in distance to the next continuous air-filled pore that were pronounced enough to impair the oxygen supply to hotspots. Air connectivity, also increased non-linearly, but did not reach a critical value (**Figure 6**), ruling out that differences in NO and $N_2O$ release at different saturations were due to gas entrapment but rather due to

445 elongated diffusion pathways in air-filled and water-filled pores, leading to longer residence times of denitrification intermediates and stronger reduction of intermediates in hotspots along the way to the headspace.

We cannot rule out bacterial spread out of the hotspots and that some denitrification might have occurred in the sand matrix. Nitrate and dissolved carbon diffused out of the hotspots and in

addition at the lowest saturation (30%WFPS) those substrates were transported convectively by capillary forces. This was demonstrated with a separate dye experiment (Figure S7). The fact that oxygen consumption was slow at 30%WFPS is indirect evidence that most bacteria remained in the hotspots. In other words, under well-aerated conditions only substrate limitation can explain a reduction in microbial growth as compared to the 60%WFPS and this can only occur if the

substrates left the hotspots, but the cells did not follow to the same degree. For the 60% and 90%WFPS cases it is unclear as to how bacterial dispersal is really relevant. *A. tumefaciens* is known to possess flagella (Merritt et al., 2007), but P. denitrificans is not motile. Experimental data on bacterial dispersal rates in soil is scarce. In a recent study with *Bacillus subtilis* and *Pseudomonas fluorescens* inoculated to repacked soils at similar bulk density (1.5g/cm³) and

saturation (60%) the first cells appeared in a distance of 15mm after nine days (Juyal et al., 2018). In our experiment, with the highest substrate concentration and the largest internal surface area in the hotspots denitrification occurred after 3-6 days. Hence, some cells may have colonized the immediate vicinity of hotspots, but cell densities outside the hotspots were likely low.

Saturation-dependent threshold behavior for denitrification is a well-studied phenomenon in soils

(Linn and Doran, 1984;Ruser et al., 2006;Paul et al., 2003), but for a lack of pore scale measurements often attributed to reduced bulk soil diffusivity. In undisturbed soil, the relative importance of air connectivity and distances between air-filled and water-filled pores might be more relevant for impairing oxygen supply and inducing denitrification. Air connectivity to the headspace was shown to affect $N_2O$ emissions in terms of magnitude and responsiveness in

repeated wetting/drying cycles in an intact soil column (Rabot et al., 2015). In agricultural soil with different crop rotations, $N_2O$ emissions were shown to correlate positively with the volume fraction of soil with macropore distances larger than 180 µm, which was used as an *ad-hoc* definition for poorly aerated soil (Kravchenko et al., 2018). In a mesocosm study on microstructural drivers for local redox conditions, none of the investigated soil pore metrics

derived from X-ray CT data (excluding those examined here) correlated with redox kinetics during a wetting/drying cycle (Wanzek et al., 2018). Hence, combining metabolic monitoring by high-resolution gas kinetics with direct assessment of diffusion lengths of gaseous and dissolved oxygen and denitrification products via X-ray microtomography emerges as a promising tool to study physical constraints for aerobic and anaerobic respiration in soil. However, meaningful

metrics derived from X-ray data relevant for denitrification are yet to be developed and will require additional experiments with both artificial and real soils. Improved understanding of factors and mechanisms controlling denitrification and N gas emission on a three-dimensional micro-scale may help to design and test soil management strategies that mediate the return of

excess nitrogen to the atmosphere in a controlled way, i.e. with as little as possible NO and $N_2O$

release, be it by crop residue (Kravchenko et al., 2017), pH (Russenes et al., 2016) or irrigation (Bergstermann et al., 2011) management. At the same time, our experiments call for the implementation of spatially explicit reaction-diffusion algorithms (Hron et al., 2015;Ebrahimi and Or, 2016) in soil process models. For instance, diffusion lengths between hotspots and air-filled pores connected to the headspace may serve as a useful metric to be implemented on model

concepts like the anaerobic soil volume fraction in larger-scale continuum models (Li et al., 2000;Schurgers et al., 2006;Blagodatsky et al., 2011).

## 5. Conclusions

Using a highly simplified model system, we demonstrate that the denitrification in heterogeneous media like soil critically depends on water saturation as well as on the spatial distribution of

microbial hotspots. Hotspot architecture effects were particularly pronounced with respect to denitrification stoichiometry resulting in vastly different NO and $N_2O$ release rates. Even though our experiment was conducted in a closed system, with growing denitrifier strains and a limited amount of substrate, the results are relevant for real soils in that they respresent a bench-mark scenario of hotspot-driven denitrification. More importantly, our reductionist approach resulted in

an experimental dataset that is amenable to process-based, pore-scale denitrification models, which will be the subject of a future study.

Hotspot architecture played a more pronounced role for denitrification kinetics at lower soil moisture (30 and 60% WFPS). Hence, denitrification and its gaseous product stoichiometry do not only depend on the amount of microbial hotspots in aerated soil, but also on their spatial

distribution. The total amount of denitrification measured as cumulative electron flow, in turn, depended more on water saturation which is in line with the well-known saturation-dependent

threshold behavior in denitrification also found in natural soil. For the case of artificial soil used in our study, we found that this threshold behavior was best explained by increased air tortuosity and air distance at high saturations. Future experiments with artificial and natural soils are needed to fully capture the regulation of denitrification at the micro-scale.

## 6. Data availability

All segmented X-ray µCT files and the GC data are permanently available through the UFZ archive with data description using standardized metadata catalogue (Dublin Core) and with access via URL: http://www.ufz.de/record/dmp/archive/7291http://www.ufz.de/record/<to-be-filled-after-acceptance>.

## 7. Author contribution

SS, HV and PD designed the experiments. SS, JZ and PD carried them out. SS, JZ and PD processed the data. SS prepared the manuscript with contributions from all co-authors.

## 8. Competing interests

The authors declare that they have no conflict of interest.

## 9. Acknowledgments

We thank Ali Ebrahimi and an anonymous reviewer for the constructive comments. This study was funded by the Deutsche Forschungsgemeinschaft through the research unit DFG-FOR 2337: Denitrification in Agricultural Soils: Integrated Control and Modelling at Various Scales (DASIM). PD received funding from the FACCE-ERA-GAS project MAGGE-pH under the Grant Agreement No. 696356. We thank Linda Bergaust for providing the bacterial strains, Jing

Zhu for laboratory support and Olaf Ippisch and Marcus Horn for helpful discussions during the planning of the experiment.

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

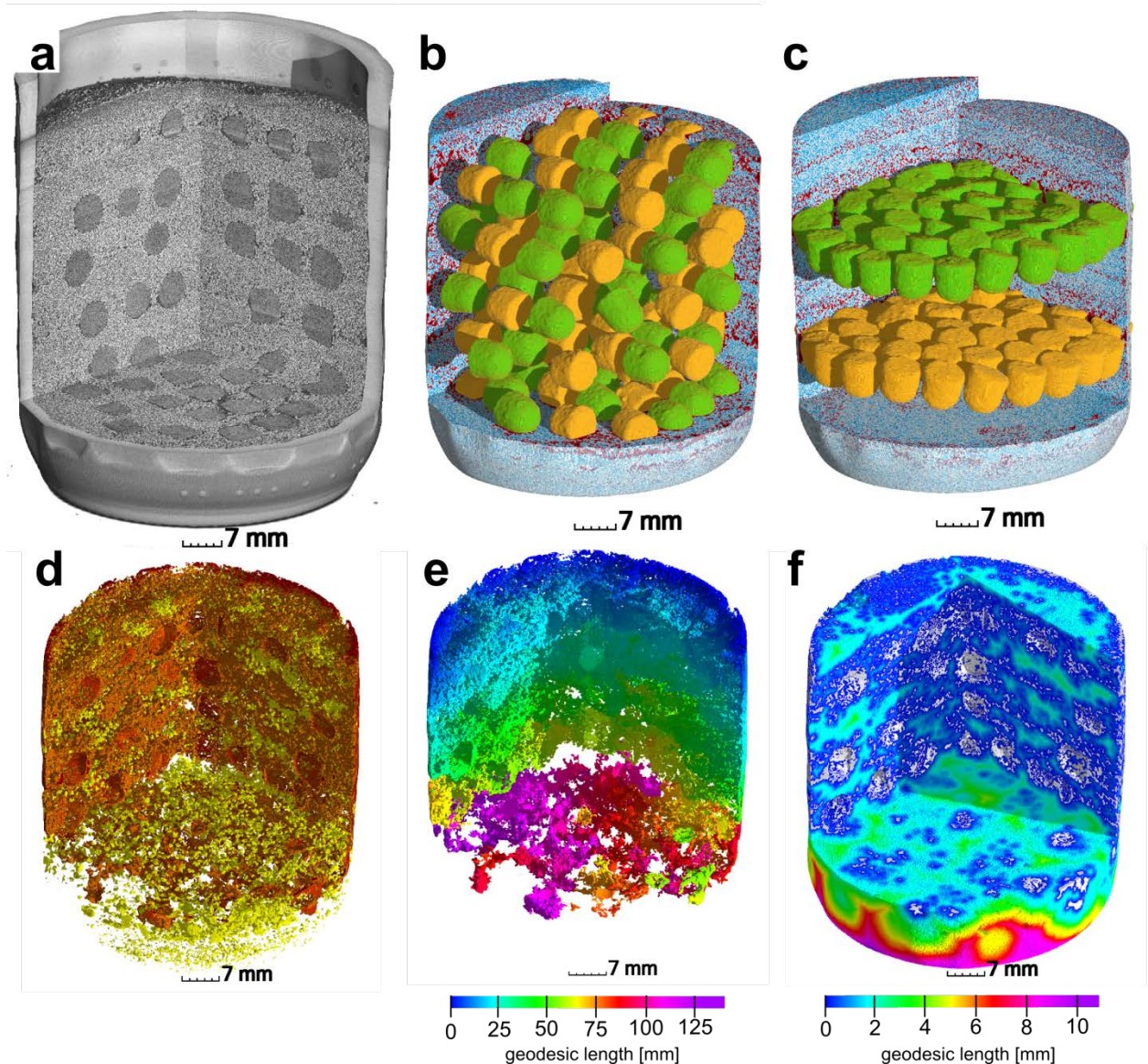

Figure 1. Upper panel: (a) X-ray CT scan of an incubation jar with random hotspot architecture and medium saturation (60% WFPS). (b) Image segmentation of the same jar into air (red), water (blue), sand (transparent), *A. tumefaciens* hotspots (orange) and *P. denitrificans* hotspots (green). (c) A different jar at medium saturation (60% WFPS) with layered *Pd/At* hotspot architecture. Lower panel: a jar with random distribution at high saturation (90% WFPS). (d) Air connectivity, determined as the volume fraction of air connected to the headspace (red, disconnected air shown in yellow). (e) Air tortuosity as derived from the geodesic length to the headspace within the connected air cluster. (f) Diffusion lengths determined as the geodesic length to the closest connected air cluster (white) within water-filled pores.

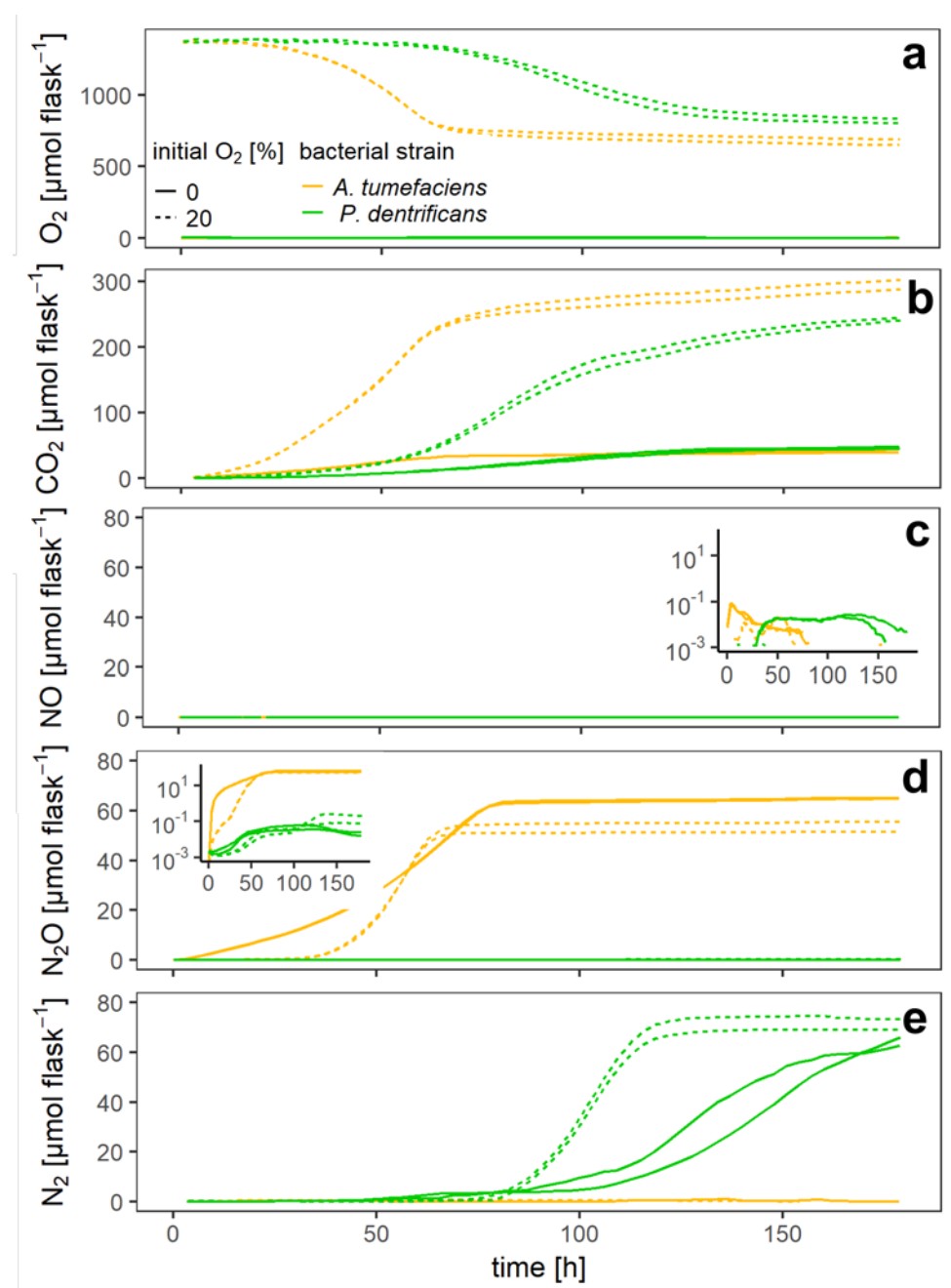

**Figure 2: Gas kinetics of individual sets of hotspots that were loosely placed in empty flasks (fifty each), saturated with a**
**growth medium and inoculated with two different bacterial strains – either a full denitrifier (*P. denitrificans*) or a**
**truncated denitrifier lacking N2O reductase (*A- tumefaciens*) - under oxic and anoxic conditions: (a) $O_2$, (b) $CO_2$, (c) NO,**
**(d) $N_2O$, (e) $N_2$.**

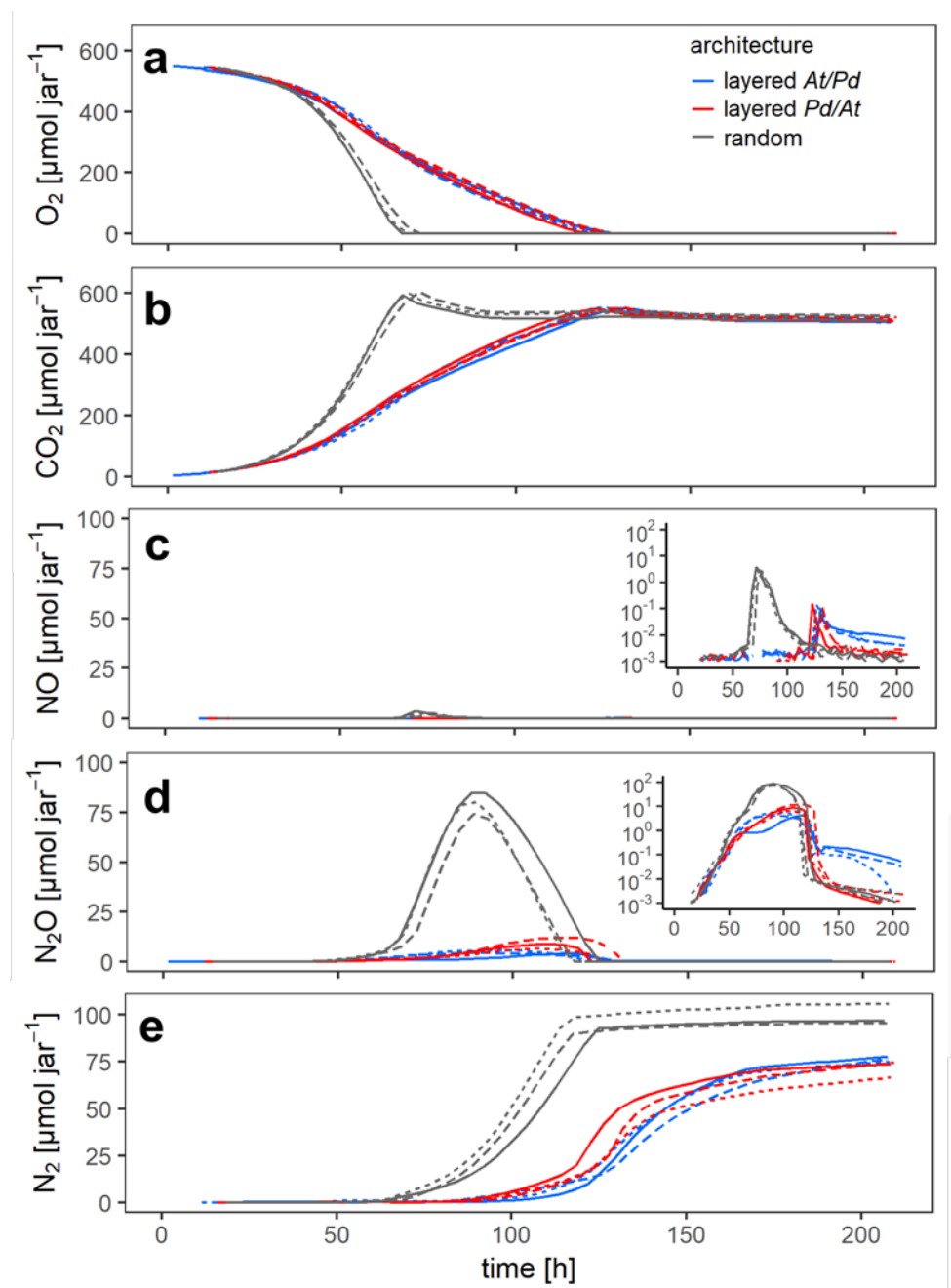

Figure 3: Gas kinetics in all treatments at medium saturation in the sand (60% WFPS) and an initial $O_2$ concentration of 20% in the headspace for three different hotspot architectures containing a total of 100 hotspots, in which the hotspots either had maximum separation distance (random) or were densely packed in layers either with all At hotspots (layered At/Pd) or all Pd hotspots (layered Pd/At) in the upper layer : (a) $O_2$, (b) $CO_2$, (c) NO, (d) $N_2O$, (e) $N_2$. Data is only shown for the first 210h of the incubation period (300h total). Different lines styles represent replicates.

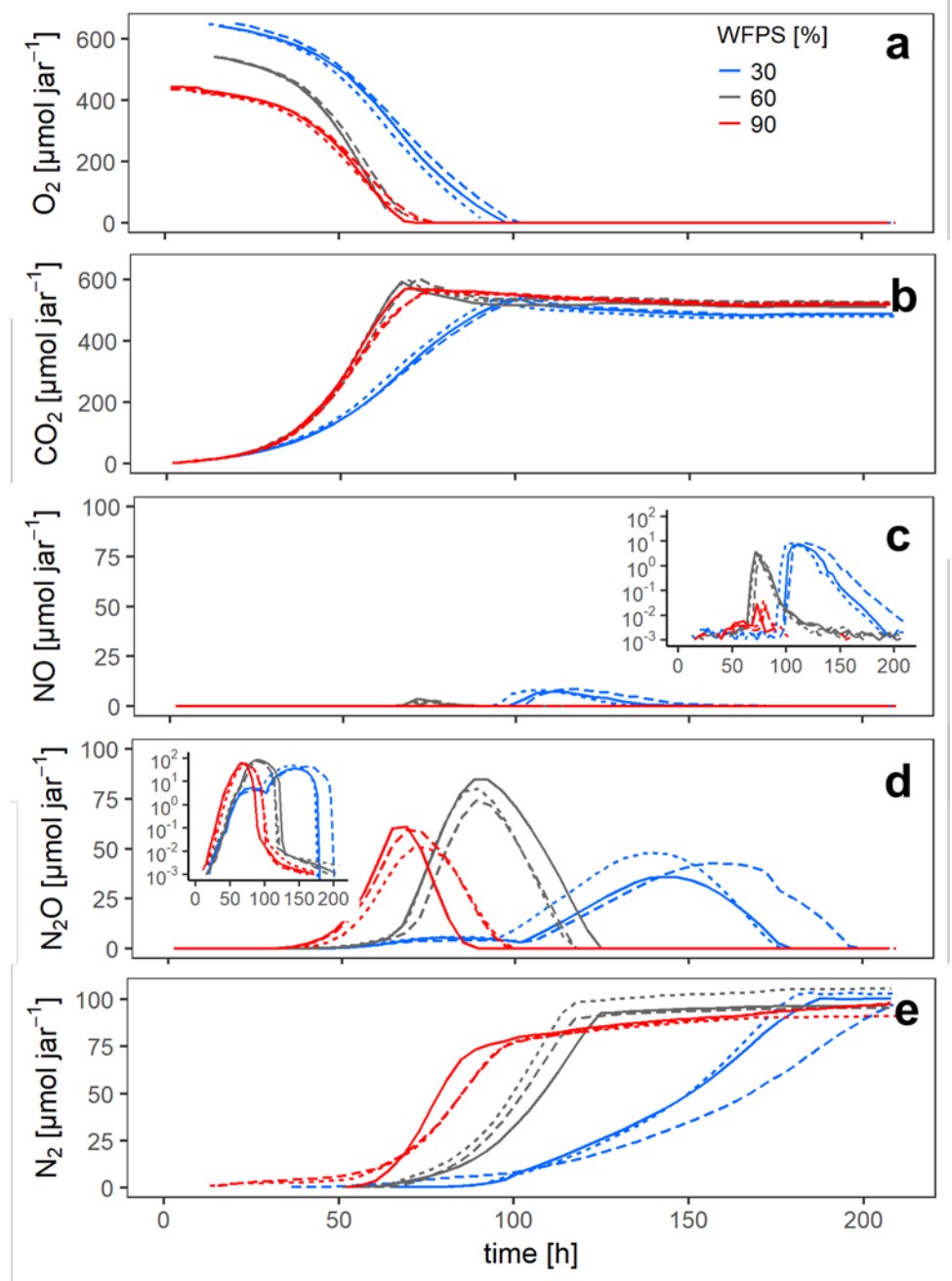

**Figure 4: Gas kinetics of randomly placed hotspots at three different saturations in the sand and an initial $O_2$**
**concentration of 20% in the headspace: (a) $O_2$, (b) $CO_2$, (c) NO, (d) $N_2O$, (e) $N_2$. Different lines styles represent replicates.**

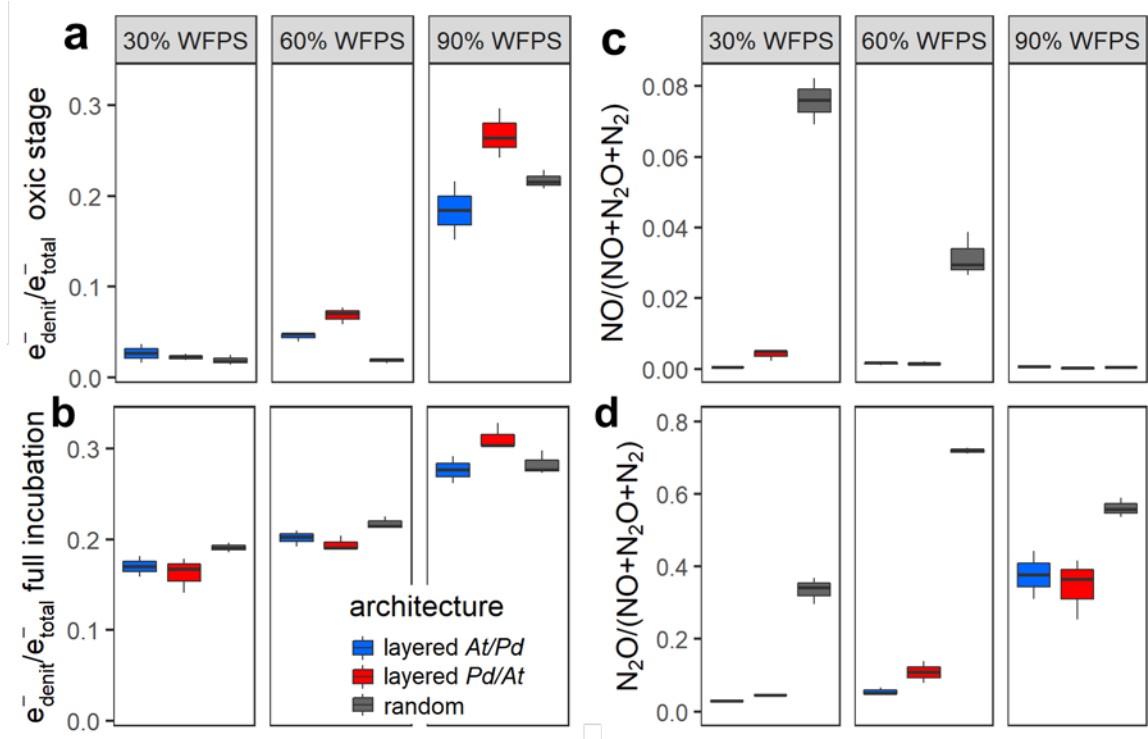

Figure 5: The proportion of denitrification in total respiration expressed as relative electron flow for all architectures and saturations. Values are reported for (a) the initial, oxic to hypoxic stage ($O_2$ present in headspace) and (b) for the full incubation period of 300 h. The product ratios for NO (c) and $N_2O$ (d) consider the full incubation period and are corrected for the release of precursor gases. Data shown as box-whisker plots: Whiskers- min-max, middle lines - median.

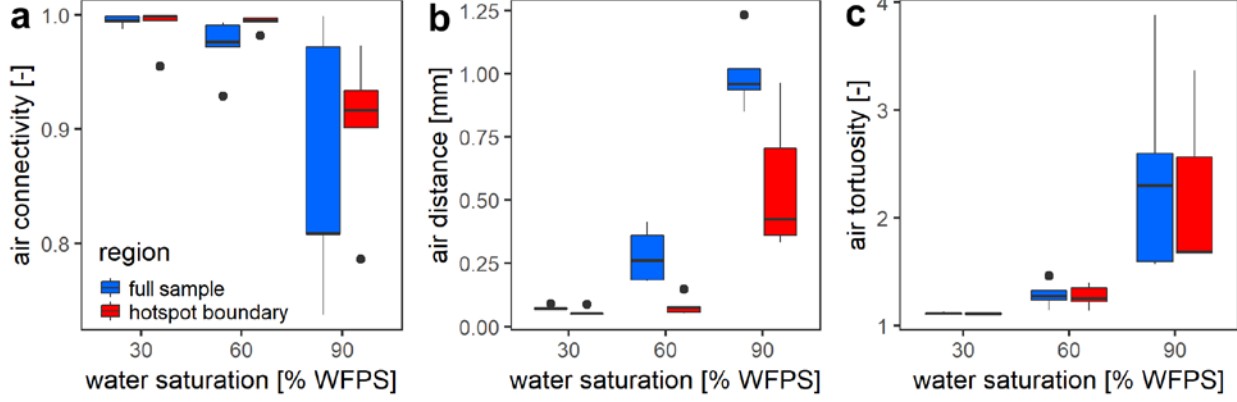

Figure 6: Morphological properties of air-filled pores at different saturations averaged over different hotspots architectures (n=5). These properties are reported separately for the entire pore space within the region of interest (full sample) and for the pore space in direct vicinity to the porous glass beads (hotspot boundary): (a) air connectivity represents the volume fraction of air with direct connection to the headspace. (b) Air tortuosity represents the ratio between geodesic length to the headspace and Euclidean distance for any voxel within the connected air-cluster. (c) Air distance represents the geodesic distance to the connected air cluster within the water-filled pores. Data shown as box-whisker plots: Whiskers- min-max, middle lines – median, dots: outliers.

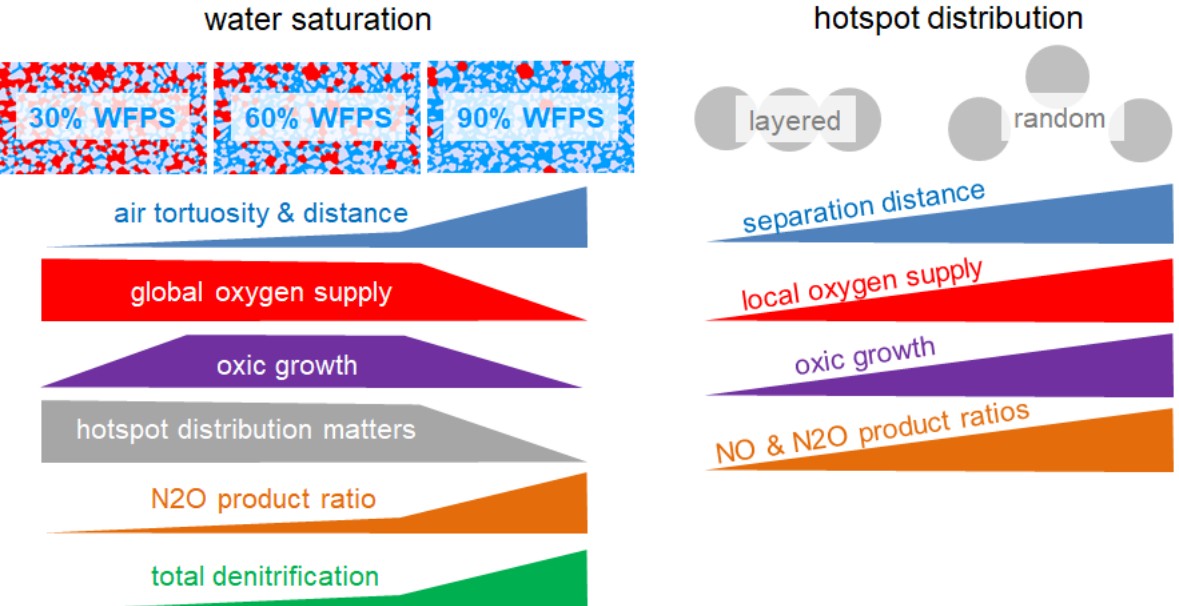

**Figure 7: Summary of the experimental findings about the impact of water saturation and spatial hotspot distribution on**
**the physical constraints on the supply of hotspots with oxygen, the resulting growth rates and the implication of various**
**aspects of denitrification.**