# Peer review of "Physical constraints for respiration in microbial hotspots in soil and their importance for denitrification"

_Biogeosciences, 2019_

## Referee Comment (RC1) · Anonymous Referee #1 · 24 Mar 2019

This is an excellent study that provides a very important insight into drivers of N2O production in soil. Systematic analysis of the hot-spots of N2O production is badly needed, and this study is a great example of how it can be done. It involves a very clever and creative experimental work, thoroughly conducted experimentation and data collection, and in-depth data analysis. The manuscript is very well written.

My only comment is that the novelty of the study and the significance of the study findings are not sufficiently highlighted in the manuscript. The conclusions, for once, do not do the due credit to the exciting results on differences between the hotspot architectures. I can see why someone might have said that this study is not novel enough by

just looking at these conclusions. To me the hotspot architecture findings are the most important and should be emphasized. For that, I would suggest to clearly describe the two architectures early-on and to rephrase the objective/hypothesis statements in the Introduction. As of now, they are rather confusing and do not clearly convey what the authors are trying to do and what they expect to see. E.g., to this reviewer it all remained rather blurry until Discussion. Some of the things that are very well put in the Discussion, e.g. lines ∼380-395 or line 400, should have made it into Introduction and Methods as hypotheses and expectation wording.

---

## Author Comment (AC1) · 26 Apr 2019

We are grateful to the reviewer for the positive feedback and constructive suggestions. In the new version of the manuscript we put more emphasis on the description of hotspot architectures already in the introduction and carve out our hypothesis more clearly that the separation distance between hotspots governs local oxygen supply which in turn regulates growth and the aerobic/anaerobic transition. In the method section we now determine the average spacing between hotspots in the random architecture (6mm) and layered architecture (1mm).

The revised paragraph in the Introduction on the objectives/hypothesis reads now as

follows:

The objective of the present study was to study the interplay between microbial activity and physical diffusion in controlling aerobic and anaerobic respiration for different spatial distributions of hotspots. We embedded uniform artificial hotspots inoculated with denitrifying pure cultures (Schlüter et al., 2018) in sterile sand in two different architectures: either densely packed in two layers with minimal distance between hotspots within a layer or distributed randomly with maximum spacing between individual hotspots. We hypothesized that the competition for oxygen would depend on this separation distance between the hotspots, which in turn would control microbial cell growth and O2 consumption and thus affect the timing of the aerobe-anaerobe transition in respiration, i.e. the onset of denitrification. Further, by placing hotspots inoculated with complete (P. denitrificans) and truncated (A. tumefaciens) denitrifiers in distinct horizontal layers instead of a signal strain in all hotspots, we expected to see interactions between these two different denitrification regulatory phenotypes with respect to overall N2O turnover. Moreover, we incubate the different hotspot architectures at three different water saturations to constrain the overall oxygen replenishment from the headspace. With this full factorial experimental setup we wanted to address the question how the overall oxygen supply (regulated by saturation) and local oxygen supply (regulated by hotspot distribution) in concert affect denitrification kinetics, total N-gas release and product stochiometry.

---

## Referee Comment (RC2) · Ali Ebrahimi (Referee) · 31 May 2019

**Summary**
The manuscript entitled "Physical constraints for respiration in microbial hotspots in soil and their importance for denitrification" by Steffen Schlüter, Jan Zawallich, Hans-Jörg Vogel and Peter Dörsch deals with studying the role of soil structure on the rates and dynamics of respiration and denitrification in a closed system. The authors manufactured porous glass beads as their model system for microbial

hotspots "soil aggregates". Porous glass beads were embedded in background of sand in a closed flasks and individual porous glass beads were incubated with either one of two soil bacterial isolates. One of the isolates, Paracoccus denitrificans perfomes the whole denitrification pathway including N2O reductase to N2 while the other isolate (Agrobacterium tumefaciens) produces N2O as the final metabolite. The idea of using controlled model systems to study the important process of denitrification in soil aggregates is highly novel and the author's attempt is clever! However, the results in the current form of the manuscript remain rather speculative and lack a meaningful transition to relevant scenarios for soil systems. At this stage, I focused on concerns regarding the design of the experiment, results and their interpretations, with minor emphasize on linguistic problems.

**Detailed review**

**Major concerns:**

I) Some of experimental decisions require better justifications in the context of denitrification process in natural soil systems. For instance, why two isolates with full or partial denitrification steps are chosen? I could think of this choice as an interesting option when studying the residence times of N2O in soil aggregates but I was confused with lack of justifications for these choices. A similar problem could be stated to the design of layered vs. random microbial hotspots "aggregates". It was nearly impossible to conceptualize how these types of distributions might actually have any relevance for soil systems or representing any specific scenarios with generalizable outcome. In the present form it is quite confusing and hard to understand what is the relevant, unknown question that the study aims to address and how the experimental design helps to navigate in that direction. The conclusions are mostly based on describing the observations and lacks discussion and interpretations to link these specific results

to advance general knowledge on soil denitrification /respiration. My understanding is that a simple mathematical model that would explain the basic mechanisms underlying the observations could go a long way to translate the results into general conclusions.

II) While authors attempted to justify the choice of closed system for their experiments, but there are fundamental issues to measure gas fluxes in the closed system. One of the main issues is that the accumulation of gases in the headspace for a long time will result into altering dissolved gas concentration profile within the soil aggregates and the soil profile itself that will ultimately affect the gas fluxes from the soil surface. From the theoretical calculations (unpublished, but shareable upon request), having even quite large headspace (1litter) would affect gas concentration profile and gas storage in soil within less than 10 hours. In other studies to resolve this issue, partial closing of the headspace is suggested (A. Ebrahimi and Or 2018). This is especially important to make sure that the oxygen concentration at the headspace remains unchanged, similar to natural soil profile throughout the experiments. In the current study, it is hard to disentangle the main mechanisms of forming anoxic microsites in the soil profile, if the main reason is because of the physical restrictions and presence of soil aggregates or just simply because of oxygen consumption and lack of oxygen in the headspace.

III) While I would like to be supportive of the idea of microbial hotspots, given the experimental design, it is hard to understand why these porous beads are hotspots for the microbial activity, compared to the full sample (background sand). First, there is no explicit observations of microbial activity at the bead scale. Second, the choice of Sistrom's medium (rich, dissolved medium) insures that substrates are uniformly distributed all over the flask (sand+beads) and knowing the fact that bacterial cells could easy spread all over the flask in the given water contents (A. N. Ebrahimi and Or 2014; Tecon and Or 2016), even if pre-inoculated only within the beads, I would then argue that the substrate and bacteria could actually be quite uniformly

distributed in the flask and turns the whole system and not only beads into "microbial hotspots". This might still be fine, if the argument of "microbial hotspot" is not based on the substrate or bacteria distribution but rather physical regulations that beads impose on the oxygen gradients, leading to anoxic hotspots in the beads. However, this argument is also unlikely given the data presented in Figure 6 that shows similar air connectivity and tortuosity for the hotspots and the full system. Surprisingly, air distance seems to be higher in the full system (Figure 6C) that could mean some regions in the full system (likely the bottom of the flask) could be even more anoxic than the porous beads. I think this aspect of the research will require better explanations of the assumptions and the reasoning behind the experimental design.

**Minor concerns:**

- In general, the Figures require more comprehensive captions. In the current form, it was painful to get full grasp of the figures without going back and forth into the text to learn the conditions that the experiments were performed.

- I think it would help a lot to use equal range for N2, NO and N2O or ploting the ratios of these gases to the total amount of available nitrogen. It was really hard to compare the rates of these gases to each other given the way the results were presented. Similar comment could be made for O2 and CO2.

- Throughout the manuscript, a few times the arguments were based on assuming that P. dentrificans is slow grower because it produces less CO2. However, this argument would only hold if both strains would have similar yield of converting substrate to biomass and CO2. Otherwise, one may argue P. dentrificans is more efficient on converting substrate to biomass and that is why produces less CO2.

- In Figure 4, scenario with 30 percent WFPS starts with about 200umol more O2 compared to 90 percent WFPS, however at the end of the experiment both scenario produces approximately 450umol CO2 with no O2 left in the flask. I was wondering where does the extra O2 is gone in 30 percent WFPS scenario? It might help to check again the mass conservations for different elements.

- I am also concerned that some of the dynamics that we see for NO and N2O gases are solely driven from the closed-nature of the experimental system. For instance, any drop in the amount of NO and N2O in the headspace observed in Figure 2 to 4 and wouldn't really happen in the open system. This type of artificial storage of reactive gases in the headspace interferes with the important storage mechanism of gases within soil aggregates (Rabot et al. 2015; Rabot, Hénault, and Cousin 2014) that significantly affect the total rate of N2O emission from soil profile (A. Ebrahimi and Or 2018).

**Recommendations:**

- While my comments may sound rather major, I still believe the study opens up a promising path toward more quantitative understanding of the denitrification process and key players in soil. I think the feature of this study is the quantification of the impact of individual factors (e.g., soil structure, water content, oxygen availability) and offering a well-controlled system with the option of disentangling multiple interacting factors. At this stage, it is fine that the experimental condition does not capture the most common scenarios in natural soil system and it would be insightful if the results would offer generalizable conclusions on the underlying mechanisms. To do this, I recommend that authors put extra work on conceptualizing the role of individual factors on the rates and patterns observed for each of gas

fluxes. I think the ideal way would be implementing a mathematical model or at minimum summarizing the results into a conceptual representation of the whole processes.

**References:**

Ebrahimi, Ali N., and Dani Or. 2014. "Microbial Dispersal in Unsaturated Porous Media: Characteristics of Motile Bacterial Cell Motions in Unsaturated Angular Pore Networks." Water Resources Research 50(9): 7406–29.

Ebrahimi, Ali, and Dani Or. 2018. "Dynamics of Soil Biogeochemical Gas Emissions Shaped by Remolded Aggregate Sizes and Carbon Configurations under Hydration Cycles." Global Change Biology 24(1): e378–92. http://dx.doi.org/10.1111/gcb.13938.

Rabot, E, C Hénault, and I Cousin. 2014. "Temporal Variability of Nitrous Oxide Emissionsby Soils as Affected by Hydric History." Soil Science Society of America Journal 78: 434–44. http://dx.doi.org/10.2136/sssaj2013.07.0311.

Rabot, E, M Lacoste, C Hénault, and I Cousin. 2015. "Using X-Ray Computed Tomography to Describe the Dynamics of Nitrous Oxide Emissions during Soil Drying." Vadose Zone Journal 14. http://dx.doi.org/10.2136/vzj2014.12.0177.

Tecon, Robin, and Dani Or. 2016. "Bacterial Flagellar Motility on Hydrated Rough Surfaces Controlled by Aqueous Film Thickness and Connectedness." Scientific Reports 6: 19409. http://dx.doi.org/10.1038/srep19409.

---

## Author Response (AR1)

**Reviewer #1 (anonymous)**

(1) **Comment**: This is an excellent study that provides a very important insight into drivers of N2O production in soil. Systematic analysis of the hot-spots of N2O production is badly needed, and this study is a great example of how it can be done. It involves a very clever and creative experimental work, thoroughly conducted experimentation and data collection, and in-depth data analysis. The manuscript is very well written. My only comment is that the novelty of the study and the significance of the study findings are not sufficiently highlighted in the manuscript. The conclusions, for once, do not do the due credit to the exciting results on differences between the hotspot architectures. I can see why someone might have said that this study is not novel enough by just looking at these conclusions. To me the hotspot architecture findings are the most important and should be emphasized. For that, I would suggest to clearly describe the two architectures early-on and to rephrase the objective/hypothesis statements in the Introduction. As of now, they are rather confusing and do not clearly convey what the authors are trying to do and what they expect to see. E.g., to this reviewer it all remained rather blurry until Discussion. Some of the things that are very well put in the Discussion, e.g. lines 380-395 or line 400, should have made it into Introduction and Methods as hypotheses and expectation wording.

(2) **Response**: We are grateful to the reviewer for the positive feedback and constructive suggestions. In the new version of the manuscript we put more emphasis on the description of hotspot architectures already in the introduction and carve out our hypothesis more clearly that the separation distance between hotspots governs local oxygen supply which in turn regulates growth and the aerobic/anaerobic transition. In the method section we now determine the average spacing between hotspots in the random architecture (6mm) and layered architecture (1mm). The revised paragraph in the Introduction on the objectives/hypothesis reads now as follows:

(3) *Changes in manuscript (line 94-114): The objective of the present study was to study the interplay between microbial activity and physical diffusion in controlling aerobic and anaerobic respiration for different spatial distributions of hotspots. We embedded uniform artificial hotspots saturated with oxically growing denitrifier pure cultures (Schlüter et al., 2018) in sterile sand in two different architectures: either densely packed in two layers with minimal distance between hotspots within a layer or distributed randomly with maximum spacing between individual hotspots (Figure 1a-b). We presumed that the competition for oxygen would depend on this separation distance between the hotspots, which in turn would control microbial cell growth and O2 consumption and thus affect the timing and velocity of the aerobe-anaerobe transition in respiration with consequences for the onset of denitrification and its gaseous product stochiometry. Further, by placing hotspots with complete (P. denitrificans) and truncated (A. tumefaciens) denitrifiers in distinct horizontal layers, we expected to see interactions between hotspots acting as mere N2O sources and hotspots potentially being N2O sinks (by reducing N2O to N2). To create contrasting velocities of oxygen transfer between headspace and hotspots, we incubated the different hotspot architectures at three water saturations thereby*

*addressing the question how overall oxygen supply (regulated by saturation) and local oxygen availability (regulated by hotspot distribution) interact in regulating denitrification activity. The overall goal of our reductionist approach with artificial hotspots, pure bacterial strains and a closed incubation system was to reduce the inherent complexity of soil in order to (i) identify processes that governed denitrification at the pore scale and (ii) create an experimental dataset for validation of pore-scale denitrification models.*

**Reviewer #2 (Ali Ebrahimi)**

We are grateful to Ali Ebrahimi for his constructive suggestions to improve the quality of the paper. In the following we reply as comprehensively as possible in individual blocks to each of the issues raised. Comments are shown in gray, Responses in black and changes to the manuscript in *italic*:

*I) Some of experimental decisions require better justifications in the context of denitrification process in natural soil systems. For instance, why two isolates with full or partial denitrification steps are chosen? I could think of this choice as an interesting option when studying the residence times of N2O in soil aggregates but I was confused with lack of justifications for these choices. A similar problem could be stated to the design of layered vs. random microbial hotspots "aggregates". It was nearly impossible to conceptualize how these types of distributions might actually have any relevance for soil systems or representing any specific scenarios with generalizable outcome. In the present form it is quite confusing and hard to understand what is the relevant, unknown question that the study aims to address and how the experimental design helps to navigate in that direction. The conclusions are mostly based on describing the observations and lacks discussion and interpretations to link these specific results to advance general knowledge on soil denitrification /respiration. My understanding is that a simple mathematical model that would explain the basic mechanisms underlying the observations could go a long way to translate the results into general conclusions.*

The rationale for using two functionally different denitrifier pure cultures is threefold: 1. The regulatory phenotypes of *P. denitrificans* and *A. tumefaciens* are well known and result in characteristic and reproducible gaseous product stoichiometries as described in section 3.1 (Lines- 200-217). Knowing growth and product kinetics in unconstrained pellets beforehand (section 3.1 - experiment without sand matrix), our aim was to deduce the activity and interaction of functionally distinct hotspots in the sand experiments from overall gas kinetics, and collate them with the experimental conditions (hotspot architecture and saturation); 2. Unlike for natural denitrifying consortia, the growth parameters of the two strains are known for a range of conditions (temperature, pH, oxidant to reductant ratios, etc) and the outcome of our experiment thus lends itself to 3D transport-reaction modelling explicit for microbial hotspots in a follow up study. 3. A reductionist approach is helpful to advance process understanding before moving to complex dynamics in real soil systems (Rillig and Antonovics, 2019). We have rewritten parts of the introduction, and hope to have clarified the rational of our experimental setup:

*Lines 103-113: Further, by placing hotspots with complete (P. denitrificans) and truncated (A. tumefaciens) denitrifiers in distinct horizontal layers, we expected to see interactions between hotspots acting as mere $N_2O$ sources and hotspots potentially being $N_2O$ sinks (by reducing $N_2O$ to $N_2$). To create contrasting velocities of oxygen transfer between headspace and hotspots, we incubated the different hotspot architectures at three water saturations thereby addressing the question how overall oxygen supply (regulated by saturation) and local oxygen availability (regulated by hotspot distribution) interact in regulating denitrification activity. The overall goal of our reductionist approach with artificial hotspots,*

*pure bacterial strains and a closed incubation system was to reduce the inherent complexity of soil in order to (i) identify processes that governed denitrification at the pore scale and (ii) create an experimental dataset for validation of pore-scale denitrification models.*

A lack of justification for the two different hotspot architectures was also mentioned by reviewer #1. Having the inoculated glass pellets as experimental unit, it was our intention to create contrasting hotspot architectures with respect to local O2 consumption and gaseous transport of NO and N2O between the functionally distinct hotspots. For this we chose a setup with minimal distance between hotspots (layered) and one with maximum distance (random) between the hotspots. In addition, the layered design gave us the option to place the functionally different strains into individual layers, with the complete denitrifier *P. denitrificans* either on top or in the bottom of the jar. This indeed enabled us to study the effect of functionally and spatially distinct hotspots on the residence time and fate of N2O (L. 390-395). We are aware that such a separation of different strains can hardly be found in natural soil. However, only such an artificial but well-defined spatial arrangement allows us to separate the effects of local gas consumption/production from those of diffusion. In this way it is possible to connect the macroscopic concept of hotspots which is around since quite a while with microscopic understanding of denitrification processes the pore scale. Our experimental design is motivated to allow for this connection following a reductionist approach. This is now explained in more detail in the Introduction section:

*Lines 94-103: The objective of the present study was to study the interplay between microbial activity and physical diffusion in controlling aerobic and anaerobic respiration for different spatial distributions of hotspots. We embedded uniform artificial hotspots saturated with oxically growing denitrifier pure cultures (Schlüter et al., 2018) in sterile sand in two different architectures: either densely packed in two layers with minimal distance between hotspots within a layer or distributed randomly with maximum spacing between individual hotspots (Figure 1a-b). We presumed that the competition for oxygen would depend on this separation distance between the hotspots, which in turn would control microbial cell growth and O2 consumption and thus affect the timing and velocity of the aerobe-anaerobe transition in respiration with consequences for the onset of denitrification and its gaseous product stochiometry.*

We agree that the overall research question the study aims to address has not been properly carved out in the original version of the paper, but will be clarified in the new version of the manuscript. The "hotspot" concept has been around for decades (since (Parkin, 1987)), and we urgently need experimental model systems of hotspots to validate pore-scale models of denitrification, which, in turn, are needed to develop NO and N2O mitigation strategies. That this is – necessarily - reductionist, we have mentioned above.

*II) While authors attempted to justify the choice of closed system for their experiments, but there are fundamental issues to measure gas fluxes in the closed system. One of the main issues is that the accumulation of gases in the headspace for a long time will result into altering dissolved gas concentration profile within the soil aggregates and the soil profile itself that will ultimately affect the gas fluxes from the soil surface. From the theoretical calculations (unpublished, but shareable upon request), having even quite large headspace (1litter) would affect gas concentration profile and gas storage in soil within less than 10 hours. In other studies to resolve this issue, partial closing of the headspace is suggested (A. Ebrahimi and Or 2018). This is especially important to make sure that the oxygen concentration at the headspace remains unchanged, similar to natural soil profile throughout the experiments. In the current study, it is hard to disentangle the main mechanisms of forming anoxic*

*microsites in the soil profile, if the main reason is because of the physical restrictions and presence of soil aggregates or just simply because of oxygen consumption and lack of oxygen in the headspace.*

It was never our intention to study respiration/denitrification kinetics at unchanged "steady-state" O2 conditions. Much to the contrary, we used batch incubation with a finite amount of O2 to gradually induce anoxia in the artificial hotspot system. All known heterotrophic denitrifiers are facultative aerobes, i.e. they thrive with O2 as electron acceptor. In the present experiment, we were particularly interested in the oxic-anoxic transition of denitrifiers, as it determines to which degree denitrification enzymes are induced. The extent to which and the sequence in which denitrification enzymes are induced in response to progressing anoxia determine denitrification enzyme kinetics and hence its product stoichiometry. This has its representation in the well-known fact that N2O emissions are episodic, driven by rain falls and/or periods of high auto- or heterotrophic O2 consumptions as elaborated on in the discussion (L. 362-374 in discussion paper). Our experiment conceptualizes this behavior by loading the hotspots with an easily degradable C medium together with growing denitrifiers and explicitly aims at studying the interplay between oxic and anoxic respiration and the effect of hotspot architecture and water saturation thereon. A welcome side effect of a static batch setup (as opposed to dynamic incubation at steady-state O2 levels) is that the former allows for sensitive detection of N2 at high temporal resolution, which appears justified given the fact that induction and functioning of N2O reductase (here in *P. denitrificans*) is instrumental for the understanding of N2O emissions. We have explicitly stated now the shortcoming of closed systems in the method section:

*Lines 184-188: This approach cannot make up for the general shortcoming of closed systems that accumulate gaseous and dissolved intermediates to high concentrations, but $N_2O$ concentrations in soil air reaching >100 ppm are reported occasionally (Risk et al., 2014;Russenes et al., 2019).*

III) While I would like to be supportive of the idea of microbial hotspots, given the experimental design, it is hard to understand why these porous beads are hotspots for the microbial activity, compared to the full sample (background sand). First, there is no explicit observations of microbial activity at the bead scale. Second, the choice of Sistrom's medium (rich, dissolved medium) insures that substrates are uniformly distributed all over the flask (sand+beads) and knowing the fact that bacterial cells could easy spread all over the flask in the given water contents (A. N. Ebrahimi and Or 2014; Tecon and Or 2016), even if pre-inoculated only within the beads, I would then argue that the substrate and bacteria could actually be quite uniformly distributed in the flask and turns the whole system and not only beads into "microbial hotspots". This might still be fine, if the argument of "microbial hotspot" is not based on the substrate or bacteria distribution but rather physical regulations that beads impose on the oxygen gradients, leading to anoxic hotspots in the beads. However, this argument is also unlikely given the data presented in Figure 6 that shows similar air connectivity and tortuosity for the hotspots and the full system. Surprisingly, air distance seems to be higher in the full system (Figure 6C) that could mean some regions in the full system (likely the bottom of the flask) could be even more anoxic than the porous beads. I think this aspect of the research will require better explanations of the assumptions and the reasoning behind the experimental design.

We are fully aware that the dissolved carbon substrate and nitrate diffuses out of the glass beads into the sand and that at the lowest saturation (30%WFPS) they have also been transported convectively by capillary forces. This was demonstrated with a separate brilliant blue experiment in the supporting information. The fact that oxygen consumption was slow at 30%WFPS is indirect evidence that the bacteria mainly remained in the hotspots. In other words, under well-aerated conditions only substrate limitation can explain a reduction in microbial growth as compared to the 60%WFPS and this can only

occur if the substrates left the hotspots (highest loss of all saturations), but the cells didn't follow to the same degree but instead were nurtured more steadily via diffusion of substrates back into the hotspots. For the 60% and 90%WFPS case we can only speculate, as to how far bacterial dispersal is really relevant. *A. tumefaciens* is known to possess flagella (Merritt et al., 2007), but P. denitrificans is not motile (personal communication with Linda Bergaust, NMBU, Norway). Experimental data on bacterial dispersal rates in soil is scarce. In a recent paper (Juyal et al., 2018), the first cells appeared in a distance of 15mm after nine days (216h) for *Bacillus subtilis* and *Pseudomonas fluorescens* in repacked soils at similar bulk density ($1.5g/cm^3$) and saturation (60%). In our experiment, with the highest substrate concentration in the hotspots and most oxic-anoxic transitions already occurring after 3-6 days, the bottom of the jar might not have been colonized that quickly, but this cannot be backed by data. We will discuss the probability of bacterial spread at high saturations in the discussion now.

*Lines 447-462: We cannot rule out bacterial spread out of the hotspots and that some denitrification might have occurred in the sand matrix. Nitrate and dissolved carbon diffused out of the hotspots and in addition at the lowest saturation (30%WFPS) those substrates were transported convectively by capillary forces. This was demonstrated with a separate dye experiment (Figure S7). The fact that oxygen consumption was slow at 30%WFPS is indirect evidence that most bacteria remained in the hotspots. In other words, under well-aerated conditions only substrate limitation can explain a reduction in microbial growth as compared to the 60%WFPS and this can only occur if the substrates left the hotspots, but the cells did not follow to the same degree. For the 60% and 90%WFPS cases it is unclear as to how bacterial dispersal is really relevant. A. tumefaciens is known to possess flagella (Merritt et al., 2007), but P. denitrificans is not motile. Experimental data on bacterial dispersal rates in soil is scarce. In a recent study with Bacillus subtilis and Pseudomonas fluorescens inoculated to repacked soils at similar bulk density ($1.5g/cm^3$) and saturation (60%) the first cells appeared in a distance of 15mm after nine days (Juyal et al., 2018). In our experiment, with the highest substrate concentration and the largest internal surface area in the hotspots denitrification occurred after 3-6 days. Hence, some cells may have colonized the immediate vicinity of hotspots, but cell densities outside the hotspots were likely low.*

*Minor concerns:*

- In general, the Figures require more comprehensive captions. In the current form, it was painful to get full grasp of the figures without going back and forth into the text to learn the conditions that the experiments were performed.

A minimum amount of information on the incubation conditions is now added to the captions.

I think it would help a lot to use equal range for N2, NO and N2O or ploting the ratios of these gases to the total amount of available nitrogen. It was really hard to compare the rates of these gases to each other given the way the results were presented. Similar comment could be made for O2 and CO2.

We have now used the same range for O2 and CO2 and the same range for NO, N2O and N2 (on linear scale with logarithmic scale as an inset).

Throughout the manuscript, a few times the arguments were based on assuming that P. dentrificans is slow grower because it produces less CO2. However, this argument would only hold if both strains would have similar yield of converting substrate to biomass and CO2. Otherwise, one may argue P. dentrificans is more efficient on converting substrate to biomass and that is why produces less CO2.

Yes, this statement is shaky, when only based on CO2. However, it can be made when also taking the consumption of oxygen into account. This has been stated in lines 201-202 of the discussion paper.

- In Figure 4, scenario with 30 percent WFPS starts with about 200umol more O2 compared to 90 percent WFPS, however at the end of the experiment both scenario produces approximately 450umol CO2 with no O2 left in the flask. I was wondering where does the extra O2 is gone in 30 percent WFPS scenario? It might help to check again the mass conservations for different elements.

The values represent the amount of gaseous oxygen in the flask. In the 90% WFPS there is less gaseous oxygen but more dissolved oxygen to start with, as more of the pore space is occupied by water. More specifically, the internal flask volume amounts to 238.3ml of which 118.3ml is occupied by the total pore volume including the headspace and the porosity of the sand. The liquid volume (including the liquid stored in glass beads) in the 90%WFPS and 30%WFPS treatments amounts to 49.6ml and 20.8ml, respectively. The CO2 mass balance for all treatments is shown in Figure 1, but not added to the paper.

[Figure]

**Figure 1: Total CO2 accumulated in each treatment at the end of incubation (n=3).**

- I am also concerned that some of the dynamics that we see for NO and N2O gases are solely driven from the closed-nature of the experimental system. For instance, any drop in the amount of NO and N2O in the headspace observed in Figure 2 to 4 and wouldn't really happen in the open system. This type of artificial storage of reactive gases in the headspace interferes with the important storage mechanism of gases within soil aggregates (Rabot et al. 2015; Rabot, Hénault, and Cousin 2014) that significantly affect the total rate of N2O emission from soil profile (A. Ebrahimi and Or 2018).

See response above; it was not the goal of our experiment to disentangle storage of N-gas intermediates in the headspace from N-gas intermediates stored in the pore space of in the pore water. That is the reason why we do not draw any general conclusions from our experiment to open soils under atmospheric conditions. In open systems, it is difficult to quantify how much of the reactive nitrogen has already been converted into N2 or it still present in the soil solution as nitrate or nitrite. Hence, there is a tradeoff between artificiality and measurement uncertainty, which we are currently not able to resolve.

Recommendations:

- While my comments may sound rather major, I still believe the study opens up a promising path toward more quantitative understanding of the denitrification process and key players in soil. I think the feature of this study is the quantification of the impact of individual factors (e.g., soil structure, water content, oxygen availability) and offering a well-controlled system with the option of disentangling multiple interacting factors. At this stage, it is fine that the experimental condition does not capture the most common scenarios in natural soil system and it would be insightful if the results would offer generalizable conclusions on the underlying mechanisms. To do this, I recommend that authors put extra work on conceptualizing the role of individual factors on the rates and patterns observed for each of gas fluxes. I think the ideal way would be implementing a mathematical model or at minimum summarizing the results into a conceptual representation of the whole processes.

We have made an attempt now to conceptualize the experimental findings with Fig. 2 and will add this to the Discussion section of the updated manuscript.

*Lines (363-367): For this, we compared different combinations of water saturation in the matrix and spatial distributions of hotspots, which led to different physical constraints for the supply of hotspots with oxygen. As a consequence, oxic growth rates differed among treatments which had various implications on denitrification as summarized in a conceptual scheme (Figure 7).*

[Figure]

**Figure 2: Conceptual representation of the experimental findings of the incubation setup with a full-factorial design of the factors water saturation and hotspot distribution**

References:

Juyal, A., Eickhorst, T., Falconer, R., Baveye, P.C., Spiers, A., Otten, W., 2018. Control of Pore Geometry in Soil Microcosms and Its Effect on the Growth and Spread of Pseudomonas and Bacillus sp. Frontiers in Environmental Science 6(73).

Merritt, P.M., Danhorn, T., Fuqua, C., 2007. Motility and chemotaxis in Agrobacterium tumefaciens surface attachment and biofilm formation. J Bacteriol 189(22), 8005-8014.

Parkin, T.B., 1987. Soil microsites as a source of denitrification variability. Soil Science Society of America Journal 51(5), 1194-1199.

Rillig, M.C., Antonovics, J., 2019. Microbial biospherics: The experimental study of ecosystem function and evolution. Proceedings of the National Academy of Sciences 116(23), 11093-11098.

---

## Author Response (AR2)

We thank the editor for suggesting some final edits to the manuscripts and will respond to them below. Please note that we have also corrected a few typos that we found during proofreading. Also, we have uploaded the data as stated in the Data availability statement and updated the URL link. The data is now permanently available under the following link:

http://www.ufz.de/record/dmp/archive/7291

Specific Comments:

*Line 100: I guess Figure 1b-c?*

Yes, we have corrected this now.

*Line 105: mainly due to the microbial cell growth (hardly measured by this work) or the surrounding conditions changes?*

It is true that we did not measure cell growth directly. Yet, with all other things being equal a higher O2 consumption rate can only be caused by higher cell numbers, so it is safe to use it as a proxy. Please note that this sentence is formulated as a hypothesis and not as a claim, so we prefer to keep the sentence as it is.